# Oxidative Stability of Green Coffee Oil (*Coffea arabica*) Microencapsulated by Spray Drying

**Miriam Granados-Vallejo [1], Hugo Espinosa-Andrews [1,*](ID),**
**Guadalupe M. Guatemala-Morales [1] (ID), Hugo Esquivel-Solis [1] and Enrique Arriola-Guevara [2,*](ID)**

[1] Tecnología Alimentaria, Biotecnología Médica y Farmacéutica, Centro de Investigación y Asistencia en Tecnología y Diseño del Estado de Jalisco, A.C., Normalistas 800, Guadalajara, Jalisco 44270, Mexico; migranados_al@ciatej.edu.mx (M.G.-V.); gguatemala@ciatej.mx (G.M.G.-M.); hesquivel@ciatej.mx (H.E.-S.)

[2] Departamento de Ingeniería Química, Centro Universitario de Ciencias Exactas e Ingenierías, Universidad de Guadalajara. Blvd. Marcelino García Barragán #1421, Esq. Calzada Olímpica. Guadalajara, Jalisco 44430, Mexico

**\*** Correspondence: hespinosa@ciatej.mx (H.E.-A.); enrique.arriola@academicos.udg.mx (E.A.-G.)

**Abstract:** In the search for oils of commercial interest that serve as new sources for the generation of cosmetic, pharmaceutical, or nutraceutical products, the green coffee beans oil (*Coffea arabica* L.) was studied. This research aimed to evaluate the oxidative stability of microencapsulated green coffee oil (*Coffea arabica*) by spray drying. The green coffee oil emulsions were produced by microfluidization using mesquite gum and octenyl succinic anhydride modified starches (OSA-starch) as wall-material. The particle size, polydispersity, and zeta potential on the microfluidized emulsions were optimized. The results showed that microfluidization had positive effects on the reduction of the emulsion droplets and the zeta potential, developing stable emulsions for both polymers. Then, the optimal microfluidization conditions were used to evaluate the impact of the spray drying conditions on the microencapsulation efficiency, morphology, and oxidation stability of the green coffee oil microcapsules under accelerated storage conditions (32% relative humidity (RH) at 25 °C). The microencapsulation efficiency was approximately 98% for both wall-materials. The morphology of the microcapsules showed spherical shapes and polydisperse sizes, a typical characteristic of spray-dried powders. The oxidative stability of the microcapsules was lower than the bulk green coffee oil (87.39 meq of $O_2$/kg of oil), reaching values of 60.83 meq of $O_2$/kg of oil for mesquite gum and 70.67 meq of $O_2$/kg of oil for OSA-starch. The microcapsules produced have good potential for the development of nutraceutical foods or cosmetic formulations with adequate stability.

**Keywords:** *Coffea arabica*; mesquite gum; OSA-starch; microfluidization; spray drying

## 1. Introduction

Coffee is one of the major industrial products; currently grown in about 80 countries of four continents, it is one of the most popular beverages in the world. The most important commercial species is *Coffea arabica* L., which provides more than 95% of the world's coffee [1]. Products derived from coffee have been long used by humankind as beverages, foods, and cosmetics. Most recently, the oils extracted by cold pressing from unroasted beans coffee were introduced to the cosmetic market with high impact due to its unique composition that shows an expressive antioxidant activity against lipid peroxidation and its activity on the skin health [2].

The green coffee oils are used in the nutraceutical and cosmetic industries due to their significant amounts of fatty acids, e.g., palmitic, stearic, oleic, and linoleic acids [2,3]. These polyunsaturated fatty acids are susceptible to oxidation, increasing the concentration of reactive oxygen species

(free radicals), producing volatile off-flavor products, and some metabolites which may exert adverse effects on human health [4,5]. Microencapsulation is a technology used to reduce the oxidative deterioration of the oils, forming an easy-to-handle microcapsules powders. Spray drying is one of the broadest microencapsulation processes for the protection of the functional ingredients [6]. However, the susceptibility to lipid oxidation depends on the lipid composition, wall material, antioxidant concentration, temperature, light, and spray drying conditions. The wall material is an essential factor that influences the protection of microencapsulated oils [7]. Modified starches, proteins, maltodextrins, and natural gums are some of the most widely used food-wall materials in spray drying [6]. Some of these wall materials have interfacial activities so that they also can promote emulsion formation and stability [7].

Mesquite gum is a natural protein galactomannan produced by the *Prosopis* spp. trees [8]. Mesquite gum possesses a small amount of protein (2.0 to 4.8%) attached to a polysaccharide backbone, which provides a high emulsifying capacity [9]. Recently, García et al. [10] produced stable nanoemulsions from fish oil using mesquite gum as an emulsifier by microfluidization. Besides, mesquite gum has been used as wall material for microencapsulation lemon essential oils by spray drying, showing high volatile oil retention, but lower oxidation stability of the essential oils [11]. On the other hand, octenyl succinic anhydride modified starches (OSA-starch) are used for microencapsulation of flavors, fats, oils, and vitamins. OSA-starches are suggested instead of gum arabic and gelatin as a microencapsulation agent due to their resistance to oxidation [12].

There are few reports on the manufacture of microcapsules of green coffee oil [2,3,13,14]. However, there is little information about the emulsion properties and the effect of the wall material on the protection of green coffee oil microcapsules. This research aimed to optimize the micro fluidization conditions (pressure and number of passes) to produce green coffee oil nanoemulsions using mesquite gum or OSA-starch as emulsifying materials. Smaller emulsions were selected to produce microcapsules of the green coffee oil by spray drying, and their oxidative stabilities were evaluated under accelerated storage conditions.

## 2. Materials and Methods

Green coffee beans (*C. arabica*) from Talpa de Allende (Jalisco, Mexico) were used to extract the green coffee oil using an extruder extractor (OilPress.Co, Mondovi, WI, USA). Mesquite gum (*Prosopis laevigata*) teardrops were acquired from Natural Products of Mexico SA de CV (Morelos, Mexico). OSA-starch (Hi-cap 100) was purchased from Ingredion (Jalisco, Mexico). Magnesium chloride ($MgCl_2$) was obtained from J.T. Baker (Jalisco, México). Hexane (anhydrous 95%) and sodium thiosulfate ($Na_2S_2O_3$) were purchased from Sigma Aldrich, Inc. (St. Louis, MO, USA). Deionized distilled water was used in all experiments.

### 2.1. Formulation of Emulsion

Mesquite gum and OSA-starch solutions (20% *w/w*) were stirred for 12 h and kept overnight to warranty the full hydration of polysaccharides. The emulsions were produced using a two-step procedure. First, two primary emulsions were produced using a high shear mixer L5M-A (Silverson Machines, Inc, East Longmeadow, MA, USA) at 5000 rpm for 10 min. Then from each biopolymer, eleven emulsions were prepared according to a $3^2$ factorial design with two central points using a microfluidizer M-110 PS (Microfluidics International Co. Newton, MA, USA). The pressure varied from 34.5 MPa to 206.8 MPa, and the number of passes (N) from 1 to 5. The microfluidizer interaction chamber was a 75 μm-diameter diamond Y type (F12Y). A tubular heat exchanger was used in-line upon exit of the interaction chamber to cool the emulsion at 20 °C. The oil volume fraction was kept constant ($\varphi = 0.1$). The droplet size, polydispersity index, and zeta potential of the emulsions were evaluated.

## 2.2. Droplet Size and Polydispersity Index

The droplet size and polydispersity index of the emulsions were evaluated using a Zetasizer Nano ZS90 equipment (Malvern Instruments, Malvern, UK). Emulsions were diluted with deionized water (1/100 *v/v*). The Stokes-Einstein equation calculated the droplet size: $d_h = k_B T / 3\pi\eta_s D$; where $k_B$ is the Boltzmann constant, T is the absolute temperature, $\eta_s$ is the dynamic viscosity of the solvent, and D is the z-average translational diffusion coefficient [10].

## 2.3. Zeta Potential

The zeta potential of the emulsions was evaluated using an electrophoretic light scattering equipment Zetasizer Nano ZS90. The emulsions were diluted with deionized water (1/100 *v/v*). The electrophoretic mobility was estimated into an electric field and then converted measurements into zeta potential values using the Smoluchowsky model. Experimental data were reported as the average of three independent measurements, and the results as the mean and standard deviation [10].

## 2.4. Microencapsulation by Spray Drying

The smaller emulsions for each biopolymer were selected to produce microcapsules by spray-dried (DL410, Yamato Scientific America Inc., Santa Clara, CA, USA). The microcapsules were produced according to the experimental design of $3^2$. The experimental factors were the inlet temperature (170 and 200 °C) and the emulsion feed rate (1.5 to 2.5 mL/min). The response variables of the microcapsules were the microencapsulation efficiency and peroxide index. The atomization air was 0.4 m$^3$/min and the air pressure was 0.4 MPa. The microcapsules were collected and stored in sealed polyethylene bags and store in the dark at 20 °C until analyses. The morphology and fatty acid profile of selected microcapsules were analyzed.

## 2.5. Microencapsulation Efficiency

Microencapsulation efficiency (*ME*) of green coffee oil was estimated according to the method described by Calvo et al. [15]. One gram of the microcapsules were dispersed in 40 mL of hexane for one min. Then, the solvent was filtered through a Whatman filter paper no. 1, and the unencapsulated oil was collected. The microcapsules on the filter were washed three times with 10 mL of hexane. The solvent was removed using a rotary vacuum evaporator (RV 8V, IKA, Germany) to obtain the surface oil mass. The microencapsulation efficiency was calculated from Equation (1):

$$ME = \frac{(\text{Total oil} - \text{Surface oil})}{(\text{Total oil})} \times 100\% \tag{1}$$

## 2.6. Peroxide Index

The microcapsules and the bulk green coffee oil were stored in a vial with a controlled water activity of 0.32 at 25 °C. The peroxide index (*PI*) from the microcapsules was determined using the AOCS method Cd 8–53 over 30 days of storage [16], calculated according to Equation (2):

$$PI = \frac{(MA - B) * N * 1000}{M} \tag{2}$$

where *MA* are mL of $Na_2S_2O_3$ spent in the titration of the sample; *B* is mL of $Na_2S_2O_3$ spent in the target titration; *N* is the normality of $Na_2S_2O_3$; *M* is mass of the sample.

## 2.7. Morphology of the Microcapsules

Microcapsules were observed by a scanning electron microscope (SEM) (MIRA3 LMU, Tescan, Brno, Czechia). The samples were fixed using a two-sided carbon adhesive tape and coated for 2 min with

gold under vacuum using SPI-Module Sputter coater (West Chester, PA, USA). The microcapsules were observed using SEM at an accelerating voltage of 10 kV with an EDS detector [17].

### 2.8. Fatty Acids Profile of the Microcapsules

HPLC was used to identify the composition of fatty acids contained in the microcapsules (linoleic, palmitic, oleic, stearic, and linolenic). The oil extract from the microcapsules was carried out to 25 °C with a controlled water activity of 0.32. First, two grams of the powders were weighed and dissolved in 15 mL of water. Then, 30 mL of hexane was added to extract the oil, water was removed by phase separation, and hexane was evaporated. Finally, the fatty acids profile (C16–C18) was determined according to the Mexican standard NMX-F-490 [18] by HPLC.

### 2.9. Statistical Analysis

Experimental data were reported as the average of three independent measurements and performed based on the factorial design. Statistical analysis of the long-term stability experiment was performed using the ANOVA test. Significant differences among different treatments were performed using Tukey and LSD test ($p < 0.05$) using Statgraphics Centurion XVI software (version 16.2.04). The optimization of multiple variables was carried out, and droplet size, polydispersity index, and zeta potential were included. The homogenization conditions were determined through the desirability ($\geq 0.8$) represented in the contours of the response surface. The selected independent variables were selected based on our previous study [10]:

$$Y \; = \; b_0 + \sum_{i=1}^{2} b_i X_i + + \sum_{i,j}^{2} b_{ij} X_i X_j + \sum_{i=1}^{2} b_{ii} X_i^2 \tag{3}$$

where, $b_0$ is the constant term of the regression equation, the coefficient $b_1$ and $b_2$ are the linear terms, $b_{11}$ and $b_{22}$ are the quadratic terms, while $b_{12}$ is the interaction term. The quality of the fit was expressed with the coefficient of the determination ($R^2$). The contour plots were used to specify the interrelationships between significant variables [10]. Multiple responses were minimized according to desirable characteristics of the nanoemulsions, i.e., small mean droplet size (<200 nm), low polydispersity index (<0.2), and low zeta potential ($<-30$ mV).

## 3. Results

### 3.1. Microfluidizer Emulsions Production

The nanoemulsions were developed using a two-step procedure. In the first step, primary emulsions were performed using a high shear mixer. Then, the primary emulsions were processed several times according to the experimental design by the microfluidizer processor. The mean droplet size of the primary OSA-starch emulsions (3.20 ± 0.15 μm) was more significant than the primary mesquite gum emulsion (2.10 ± 0.17 μm). In the second step, the mean droplet size of the emulsions decreased by high disrupted forces (shear, collision, and cavitation) produced in the microfluidizer chamber [19]. The final droplet size distribution is related to the capacity of the emulsifier agent to stabilize the new interfaces produced during the size reduction. Jafari et al. [20] suggested that the final droplet size is the result of a delicate balance between the droplet disruption and the droplet coalescence. The mean droplet sizes were reduced between 334 and 153 nm using mesquite gum, while the OSA-starch reduced the mean droplet sizes between 297 and 134 nm (Table 1). These results are different from those reported in the literature. Silva et al. [13] reported that the mean droplet sizes of green coffee oil emulsions produced with a high-pressure homogenizer (50 MPa) using different polysaccharides (Hi-Cap, Capsul, N-Lok, gum Arabic, and MD10) were approximately 1.05 and 1.51 μm. García et al. [10] found that mesquite gum produced fish oil nanoemulsions approximately between 250 and 150 nm, after one to five passes of microfluidization. The results showed that the emulsion droplet sizes were influenced by the emulsifier agent, type of oil, and method of production. The polydispersity of an emulsion

represents the fraction of particles in different size classes [21]. The mesquite gum produced low polydispersity emulsions showing values of the polydispersity index from 0.071 ± 0.02 to 0.165 ± 0.02, while OSA-starch produced more polydisperse emulsions showing values from 0.098 ± 0.01 to 0.400 ± 0.02 (Table 1). All emulsions showed monomodal distributions after five passes of microfluidization.

**Table 1.** The droplet of size, polydispersity index, and zeta potential for mesquite gum and OSA-starch emulsions according to the factorial design.

| Pressure (MPa) | Number of Passes | Droplet of Size (nm) | | Polydispersity Index | | Zeta Potential (mV) | |
|---|---|---|---|---|---|---|---|
| | | Mesquite gum | OSA-Starch | Mesquite gum | OSA-Starch | Mesquite gum | OSA-Starch |
| 34.5 | 1 | 334.0 ± 7.86 [a] | 297.4 ± 9.18 [a] | 0.134 ± 0.03 [a,b] | 0.163 ± 0.01 [a,c] | −36.8 ± 0.36 [a] | −23.0 ± 0.26 [a] |
| 34.5 | 3 | 289.9 ± 8.16 [b] | 237.2 ± 5.58 [b] | 0.101 ± 0.03 [a,b] | 0.119 ± 0.01 [a,b] | −36.2 ± 0.45 [a,b] | −23.9 ± 3.05 [a,b] |
| 34.5 | 5 | 277.9 ± 7.57 [b,f] | 216.5 ± 8.82 [c,d] | 0.072 ± 0.02 [a] | 0.098 ± 0.01 [b] | −37.3 ± 0.21 [a] | −23.9 ± 0.10 [a,b] |
| 120.6 | 1 | 259.8 ± 8.13 [f] | 220.8 ± 1.01 [c] | 0.165 ± 0.02 [b] | 0.187 ± 0.02 [c,d] | −35.8 ± 2.30 [a,b,c] | −19.7 ± 0.35 [c,d,e] |
| 120.6 | 3 | 191.9 ± 6.41 [e] | 162.9 ± 0.20 [f] | 0.135 ± 0.03 [a,b] | 0.179 ± 0.01 [c,d] | −36.3 ± 1.08 [a,b] | −21.4 ± 0.51 [a,d] |
| 120.6 | 3 | 191.8 ± 4.21 [e] | 162.0 ± 4.42 [f] | 0.144 ± 0.02 [a,b] | 0.182 ± 0.01 [c,d] | −35.5 ± 0.35 [a,b,c] | −21.1 ± 0.76 [a,d,e] |
| 120.6 | 3 | 194.4 ± 5.92 [e] | 164.8 ± 4.06 [f] | 0.108 ± 0.05 [a,b] | 0.169 ± 0.02 [a,c,d] | −29.9 ± 2.82 [d] | −18.9 ± 0.17 [c,d,e] |
| 120.6 | 5 | 169.9 ± 1.15 [d,g] | 137.8 ± 1.86 [e,g] | 0.120 ± 0.01 [a,b] | 0.163 ± 0.03 [a,c] | −38.4 ± 0.52 [a] | −18.2 ± 0.35 [c] |
| 206.8 | 1 | 239.2 ± 5.53 [c] | 203.8 ± 3.09 [d] | 0.151 ± 0.03 [a,b] | 0.180 ± 0.02 [c,d] | −32.1 ± 0.53 [b,c,d] | −26.2 ± 0.32 [b] |
| 206.8 | 3 | 172.5 ± 8.03 [e] | 133.8 ± 3.25 [e] | 0.126 ± 0.02 [a,b] | 0.218 ± 0.01 [d] | −31.7 ± 0.93 [c,d] | −26.5 ± 0.21 [b] |
| 206.8 | 5 | 153.4 ± 2.24 [d] | 151.1 ± 4.18 [f,g] | 0.096 ± 0.01 [a,b] | 0.400 ± 0.02 [e] | −27.8 ± 2.89 [d] | −18.4 ± 0.40 [c,e] |
| – | – | 2103.7 ± 169.33 | 3207.2 ± 152.75 | 0.142 ± 0.12 | 0.408 ± 0.19 | −36.8 ± 0.26 | −6.7 ± 0.23 |

a, b, c, d, e, f, and g The average values (±SD, n = 3) with different letters in the columns mean statistical differences (*p* < 0.05) according to the Tukey test.

In general, the mean droplet size and polydispersity of the emulsions produced with mesquite gum decreased with increasing the pressure and the number of passes of microfluidization, enhancing the stability of the emulsions. The enhancing stability was attributed to the excellent surface properties of mesquite gum, that contains a small fraction of protein in its structure, which is absorbed in the interface of the oil droplets, while the carbohydrate fraction is spoused out to the aqueous phase, giving it a stabilization by steric and electrostatic effects [22]. On the other hand, the mean droplet size and polydispersity of the emulsions produced with OSA-starch decreased with the number of passes of microfluidization at low and moderate pressure. However, when a pressure of 208.8 MPa was used, the mean droplet size and polydispersity of the OSA-starch emulsions decreased from one to three passes and then increased from 3 to 5 passes (Table 1). This phenomenon is known as "overprocessing" of the emulsion [19,22]. The overprocessing of the OSA-starch emulsions produced a decreasing their stability during storage. Several physicochemical mechanisms have been proposed to understanding the emulsion overprocessing [23–25]: (1) a relatively slow rate of emulsifier absorption compared to the coalescence rate, (2) a relatively short residence time of droplets within the disruptive zone, (3) depletion of emulsifier, (4) loss of the emulsifier functionality to stabilize the new interfaces due to the considerable increase of the surface area or changes in the physicochemical properties of the emulsion.

Zeta potential is a measure of the emulsion charge density and is employed as a signal of the stability of the emulsion by an electrostatic effect [26]. The zeta potential of the microfluidized emulsions was significantly more negative than the primary emulsions. The negative zeta potential of the emulsions is related to the anionic nature of the carboxylic and hydroxylic groups of the mesquite gum and OSA-starch [27] and the dispersed medium conditions (pH, ionic strength, and temperature) [23]. Mesquite gum is a suitable emulsifier and stabilizer agent since its protein fraction reduces the surface tension in a liquid–liquid interface. Besides, their glucuronic acid moieties are dissociated, forming a polyvalent macroion with a large number of counter-charge counterions, maximizing the repulsive forces between them [28]. In general terms, the zeta potential of the emulsions produced with mesquite gum was not impacted by the increase in the number of passes of microfluidization from one to five. On the other hand, the OSA-starch contains hydrophobic and hydrophilic groups improving their emulsifying properties due to the incorporation of alkenyl hydrophobic groups in a normally

hydrophilic starch molecule, acquiring active surface properties which are useful for stabilizing oil-in-water emulsions [29]. The zeta potential values of the OSA-starch emulsions were lesser than the mesquite gum emulsions. Colloidal particles are considered stable to aggregation if their zeta potential is more positive than +30 mV or more negative than −30 mV since repulsive forces between droplets are predominant [26]. This agrees with the results obtained in this investigation. Salvia et al. [30] evaluated the effect of the microfluidization parameters (pressure and number of passes) on the zeta potential of lemongrass oil-alginate emulsions. They observed a significant reduction of the zeta potential regardless of the pressure of microfluidization applied. García et al. [10] reported that the zeta potential of the fish oil nanoemulsions stabilized with the mesquite gum was more negative after the microfluidization processes, founding a small change of the zeta potential values with the increase of the number of passes and pressure of microfluidization. Li et al. [31] reported that the zeta potential values of lycopene emulsions produced with OSA-starch after five passes of microfluidization were found between −35 and −22 mV, favoring the stability of the emulsions.

Surface response methodology was applied to know the effect of the microfluidization parameters on the droplet size and polydispersity index of the emulsions (Table 1). Significant factors were selected based on the *p*-value, with a 95% confidence level. The mathematical models and the $R^2$ values are shown in Table 2. The experimental data for mean droplet size reduction from green coffee oil emulsion stabilized with mesquite gum fitted well with the predicted value of the model. The standard deviation of the model was 5.43. All variables have significant effects on the mean droplet size reduction from green coffee oil emulsion. The pressure of homogenization was found to have a significant effect on the mean droplet size reduction in comparison to other variables. The number of passes and the quadratic effect of the pressure significantly reduced the polydispersity index of the emulsion. The pressure of the microfluidization showed little influence on the mean droplet size ($p < 0.05$). The pressure of the microfluidization significantly impacted the zeta potential value from the green coffee oil emulsion stabilized with mesquite gum. The conditions of microfluidization to produce small droplet sizes were the pressure of 121.6 MPa and four passes of microfluidization for mesquite gum (Figure 1A) producing mean droplet sizes of 187.4 ± 1.46 nm with a high negative charge −34.0 ± 0.23 mV.

The pressure, the number of passes, and the quadratic effect of the pressure were significant variables for the mean droplet size reduction for OSA-starch emulsions. Also, the quadratic effect of the number of passes showed little influence on the mean droplet size ($p < 0.06$). The polydispersity index was negatively influenced by the pressure and the number of passes, while the interaction pressure and the number of passes influenced positively. However, the coefficient values showed that the overall effect of increasing the pressure and number of passes leads to an increase in the polydispersity of the emulsion. Finally, the zeta potential of the OSA-starch emulsion had a negative impact with the number of passes and the quadratic effect of the pressure, while the pressure and the interaction effect of the pressure-number of passes and the quadratic effect of the number of passes showed a positive effect on the zeta potential values. An OSA-starch emulsion of the narrow droplet size distribution (189.2 ± 0.70 nm) and zeta potential value (−27.46 ± 2.14 mV) was obtained at an optimal condition of 206.8 MPa and three passes of microfluidization (Figure 1B). Fernandes et al. [32] found that the physical properties of the emulsion before spray drying were critical to the microencapsulation process.

**Table 2.** Optimal models predicted by the surface response methodology for green coffee oil emulsions.

| Model | Emulsifier Agent | Equation | $R^2$ |
|---|---|---|---|
| Mean droplet size | OSA-starch | $d_h = 354.5 - 1.28P - 44.4N + 3.48 \times 10^{-3}P^2 + 4.89N^2$ | 0.9452 |
| Polydispersity index | OSA-starch | $Pdi = 0.21 - 4.32 \times 10^{-4}P - 3.89 \times 10^{-2}N + 4.13 \times 10^{-4}PN$ | 0.8640 |
| Zeta potential | OSA-starch | $\zeta = -22.24 + 0.10P - 3.74N - 5.80 \times 10^{-4}P^2 + 1.26 \times 10^{-2}PN + 4.86 \times 10^{-1}N^2$ | 0.8871 |
| Mean droplet size | *Mesquite gum* | $d_h = 434.58 - 1.74P - 45.55N + 5.04 \times 10^{-3}P^2 - 4.31 \times 10^{-2}PN + 5.24N^2$ | 0.9957 |
| Polydispersity index | *Mesquite gum* | $Pdi = 1.18 \times 10^{-1} + 8.20 \times 10^{-4}P - 1.36 \times 10^{-2}N - 2.86 \times 10^{-6}*P^2$ | 0.8641 |
| Zeta potential | *Mesquite gum* | $\zeta = -38.71 + 3.62 \times 10^{-4}P$ | 0.5001 |

Where $P$ is the microfluidization pressure, $N$ is the number of passes, $d_h$ is the droplet size, $Pdi$ is the polydispersity index and $\zeta$ is the Zeta potential.

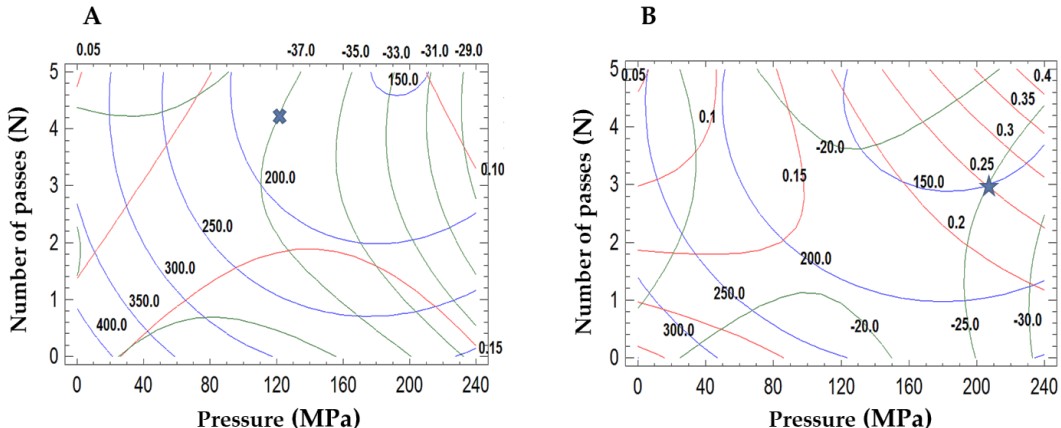

**Figure 1.** Response surface contour plots of the effects of the pressure and number of passes of microfluidization. (**A**) Mesquite gum emulsions, and (**B**) OSA-starch emulsions. Line red (–) polydispersity index, line blue (–) mean droplet size, and line green (–) zeta potential. Crossmark and star are the optimized conditions.

### 3.2. Microencapsulation of the Green Coffee Oil

Microcapsules of green coffee oil were produced by spray drying from the emulsions obtained at the optimal microfluidization conditions for each wall-material. A decrease in microencapsulation efficiency was observed with the increase in temperature when using mesquite gum, while when using OSA-starch, there are no significant differences with the increase in temperature or feed rate (Table 3). This could be related to the fact that higher inlet air temperatures affect the balance between the evaporation rate of the water and the formation of the film [33]. The elevated temperatures promote faster drying of the outer surface, compared to the internal one, which can generate cracks in the wall of the microcapsules, and this results in the release of oil. In some cases, the increase in inlet air temperature can cause heat damage to the product, with imperfections of droplet and surface growth, which increase losses during spray drying [34]. Consequently, the temperature of the inlet air is a crucial factor to consider in any microencapsulation system, to have a final product with excellent quality, and thus avoid the presence of oil on the surface which can cause losses of volatile substances, in the case of essential oils and flavors, and promote the oxidation of unsaturated fatty acids [35].

Similar results were observed from both wall materials, independently of the spray-drying conditions. These were attributed to the small droplet size of the emulsions produced by the microfluidizer, leading the high microencapsulation efficiency (approximately 98%). The encapsulation agent impacted the spray drying conditions that produced the highest yield of encapsulation. When mesquite gum was used as a wall material, an inlet temperature of 170 °C and a feed rate of 2.5 mL/min

produced a yield of encapsulation of 99.60 ± 0.10. On the other hand, when the OSA-starch was used, an inlet temperature of 200 °C and a feed rate of 1.5 mL/min produced a yield of encapsulation of 99.0 ± 0.26. Larger droplets break down during atomization in the spray-drying process, lead to higher surface oil in the microcapsules, whereas fine droplets are found to be intact during atomization. This phenomenon was observed with flaxseed oil encapsulated with gum arabic [36], and rosemary essential oils encapsulated with protein isolate and inulin [32].

**Table 3.** Microencapsulation efficiency of the green coffee oil according to the experimental design.

| Temperature (°C) | Feed Rate (mL/min) | Microencapsulation Efficiency (%) | |
| --- | --- | --- | --- |
| | | **Mesquite gum** | **OSA-Starch** |
| 170 | 1.5 | 99.4 ± 0.146 [a,b] | 98.8 ± 0.093 [a] |
| 170 | 2.0 | 99.2 ± 0.122 [a,b,c] | 98.7 ± 0.217 [a] |
| 170 | 2.5 | 99.6 ± 0.103 [a] | 98.9 ± 0.288 [a] |
| 185 | 1.5 | 99.0 ± 0.126 [b,c,d] | 98.5 ± 0.261 [a] |
| 185 | 2.0 | 99.1 ± 0.156 [a,b,c,d] | 98.7 ± 0.288 [a] |
| 185 | 2.5 | 98.8 ± 0.136 [c,d] | 98.9 ± 0.419 [a] |
| 200 | 1.5 | 98.6 ± 0.156 [d] | 99.0 ± 0.264 [a] |
| 200 | 2.0 | 98.7 ± 0.148 [d] | 98.9 ± 0.097 [a] |
| 200 | 2.5 | 98.6 ± 0.107 [c,d] | 98.8 ± 0.126 [a] |

[a, b, c, and d] The average values (± SD, n = 3) with different letters in the columns mean statistical differences ($p < 0.05$) according to the Tukey test.

OSA-starches have been used for many years for microencapsulation and beverage emulsions [27]. Similar results were reported by Carneiro et al. [37], who found that the microencapsulation efficiency values of flaxseed oil varied from 62.3% to 95.7%, being the highest value obtained for Hicap:maltodextrin. Silva et al. [13] and Carvalho et al. [14] found that the microcapsules produced with OSA-starch showed higher oil retention compared with other materials, such as gum arabic and whey protein.

### 3.3. Oxidative Stability of the Microcapsules

The surface area of the smaller particles exhibits a higher susceptibility to oxidation, once oxygen has penetrated it [38]. The green coffee oil had a peroxide index of 2.48 meq $O_2$/kg of oil. Samples showed a low oxidation level after one day of storage, ranging from 3.98 to 4.98 meq $O_2$/kg of oil. No significant difference was observed in the microencapsulated oil with mesquite gum and OSA-starch, showing that the spray drying process had a little impact on lipid oxidation of the green coffee oil. After 30 days of storage, the oxidation values of the microcapsules produced with OSA-starch were higher than the microcapsules of mesquite gum. Microcapsules produced with mesquite gum reached a peroxide index of 60.83 meq of $O_2$/kg of oil, while microcapsules produced with OSA-starch underwent a significant increase in peroxide index of 70.67 meq of $O_2$/kg of oil (Table 4).

The oxidation of microencapsulated oil is impacted by the superficial oil content, accessible to the extraction using organic solvents and the microencapsulated oil content since, for its extraction, it requires the rupture of the wall material that protects it [39]. Accordingly, the 30 days of storage produced an increase in the porosity of the microcapsules exposing the oil to oxygen; therefore, their oxidation level increased.

**Table 4.** Peroxide index in the microcapsules obtained with mesquite gum and OSA-starch.

| Spray Drying Conditions | | Peroxide Index (meq $O_2$/Kg oil) | | | |
|---|---|---|---|---|---|
| | | Mesquite gum 32% RH (Days) | | OSA-Starch 32% RH (Days) | |
| (°C) | (mL/min) | 1 | 30 | 1 | 30 |
| 170 | 1.5 | 3.99 ±0.001 [a] | 50.64 ± 1.454 [a,b] | 4.98 ± 1.410 [a] | 62.68 ± 1.398 [b] |
| 170 | 2.0 | 4.97 ± 1.399 [a] | 50.59 ± 1.439 [a,b] | 4.97 ± 1.410 [a] | 57.79 ± 0.008 [a] |
| 170 | 2.5 | 3.98 ± 0.011 [a] | 52.79 ± 1.423 [b,c] | 3.98 ± 0.003 [a] | 60.83 ± 1.376 [b] |
| 185 | 1.5 | 3.98 ± 0.002 [a] | 54.83 ± 1.371 [c,d] | 3.98 ± 0.004 [a] | 67.50 ± 0.019 [c] |
| 185 | 2.0 | 4.98 ± 1.407 [a] | 49.82 ± 0.028 [a] | 4.98 ± 1.416 [a] | 67.53 ± 0.057 [c] |
| 185 | 2.5 | 3.99 ± 0.003 [a] | 55.70 ± 0.016 [d,e] | 3.98 ± 0.009 [a] | 68.37 ± 1.382 [c,d] |
| 200 | 1.5 | 3.98 ± 0.010 [a] | 57.49 ± 0.040 [e,f] | 4.97 ± 1.409 [a] | 68.69 ± 1.398 [c,d] |
| 200 | 2.0 | 4.98 ± 1.412 [a] | 60.83 ± 1.376 [g] | 4.98 ± 1.397 [a] | 70.67 ± 1.378 [d] |
| 200 | 2.5 | 4.97 ± 1.407 [a] | 58.61 ± 1.446 [f,g] | 4.98 ± 1.404 [a] | 69.64 ± 0.039 [c,d] |

[a, b, c, d, e, f, and g] The average values (±SD, n = 3) with different letters in the columns mean statistical differences ($p < 0.05$) according to the LSD test.

The peroxide index obtained from the bulk green coffee oil was of 87.39 ± 2.759 meq of $O_2$/kg of oil after 30 days of storage. This demonstrates that the microencapsulated oil was less susceptible to the oxidation process and confirms the protective effect provided by the microencapsulation process. These results agree with those reported in other studies. Partanen et al. [40] evaluated the effect of storage conditions on the oxidation stability of microencapsulated flaxseed oil by spray drying using whey protein as a wall material. As in the present work, the authors also found that lipid oxidation was lower in microencapsulated samples compared with bulk oil. Carneiro et al. [37] evaluated the potential of the combination of maltodextrin with four types of wall materials (gum arabic, whey protein concentrate, and two types of modified starches) as alternative materials for the microencapsulation of flaxseed oil by spray drying. The bulk oil showed the highest concentration of peroxide after four weeks, reaching values of approximately 120 and 140 meq of $O_2$/kg of oil, respectively. These values are higher than those reported in this study, probably because of the unsaturation content of flaxseed oil.

*3.4. Morphology of the Microparticles*

SEM images showed the surface morphology of selected microcapsules. The microcapsules exhibited spherical shapes of several sizes with outer surfaces free of pores, common characteristics of polysaccharide spray-dried powders. Furthermore, the lack of wall fissures or porosity on the particle surface displayed a complete coverage of the wall material over the core. The most evident structural difference between the samples was of surface topology, specifically the presence of depressions. Microcapsules produced with mesquite gum did not show fissures, but more significant dents were observed on the surface to those obtained with OSA-starch (Figure 2), which a few microcapsules showed a spherical shape with no cracks or holes. Xie et al. [41] observed that the morphology of microcapsules of vitamin A produced by spray drying was different, the particles produced with OSA-starch (Hi-Cap 100) had a smoother surface than the particles constituted with gelatin-sucrose and gelatin-peach-gum-sucrose.

The microstructure of the green coffee oil microcapsules after 30 days of storage at 32% RH are shown in Figure 3. The microcapsules produced with mesquite gum did not show visual changes in the morphologies. However, OSA-starch microcapsules presented some particles with cracks, fissures, and with holes. These changes can promote the exposition of green coffee oil with environmental conditions (light, oxygen, humidity), accelerating the oxidation process [13]. Shamaei et al. [42] found that damages on the surfaces of microcapsules might contribute to the diffusion of oil droplets to the microcapsules surfaces. This behavior agrees with the oxidative stability results, where the microcapsules stored in the 32% RH showed a higher peroxide index.

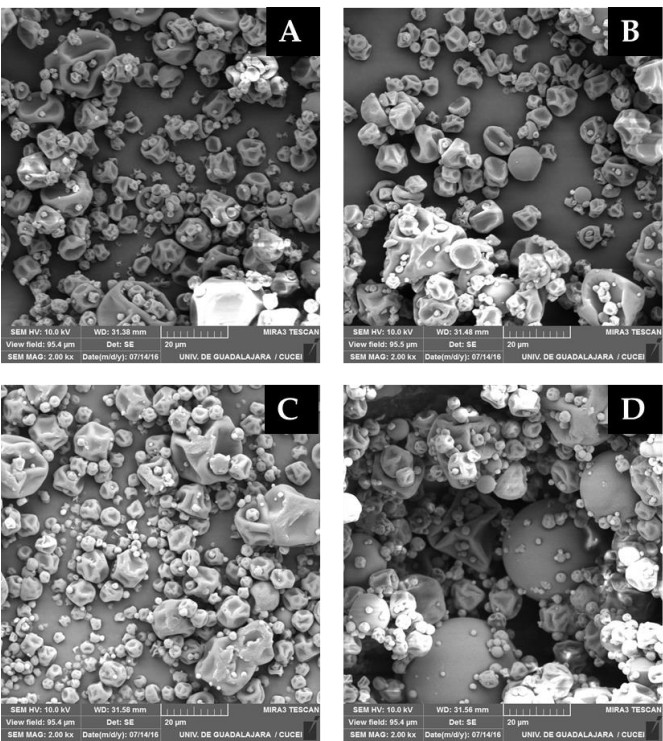

**Figure 2.** SEM images of spray-dried microcapsules of the green coffee oil. (**A**,**B**) Mesquite gum microcapsules produced at 185 °C (left) and 200 °C (right), and (**C**,**D**) OSA-starch microcapsules produced at 170 °C (left) and 200 °C (right).

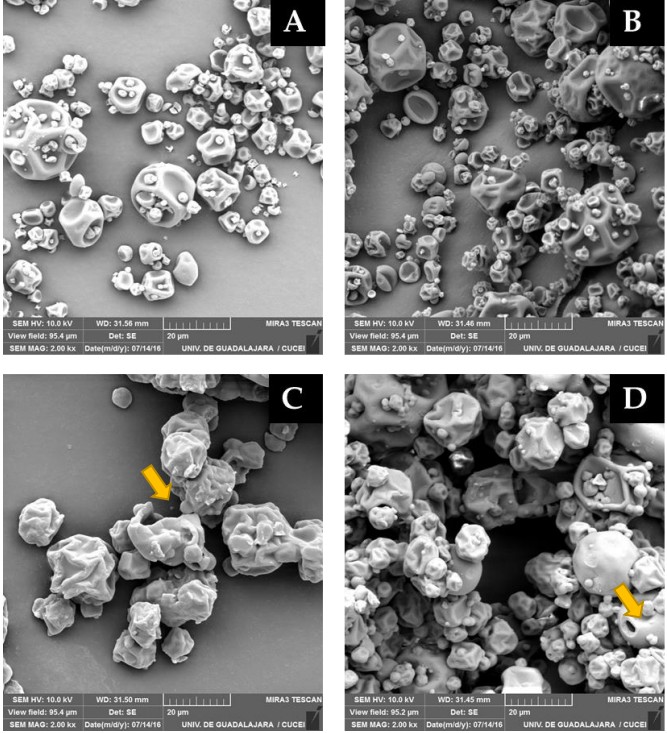

**Figure 3.** SEM images of spray-dried microcapsules of green coffee oil after 30 days of storage 32% RH. (**A**,**B**) Mesquite gum microcapsules produced at 185 °C (left) and 200 °C (right), emulsion stabilized by mesquite gum, and (**C**,**D**) OSA-starch microcapsules produced at 170 °C (left) and 200 °C (right).

### 3.5. Fatty Acid Profile of the Microcapsules

All samples were characterized by a high concentration of polyunsaturated fatty acids that constitute approximately 45%, of which the most abundant was linoleic acid. Fatty acid profiles remained similar after 30 days of storage 32% RH concerning of storage of bulk green coffee oil, despite showing different peroxide indexes between one day and 30 days of storage, probably due to the presence of traces of chlorogenic acid in the samples, since it gives them an antioxidant quality that counteracts the oxidation produced mainly by free radicals, chemicals, light, and moisture [1]. However, it should be kept in mind that the oxidative stability of oils is influenced by the amount and type of natural antioxidants, phospholipids, free fatty acids, mono- and di-glycerides, polymers, and the number of double bonds in the oil [43]. According to the initial composition of the fatty acids, it was observed that green coffee oil that had not been microencapsulated was more susceptible to oxidation reactions. In the bulk green coffee oil, changes were observed mainly in the composition of linoleic acid, which decreased to a value of 9.97%, this fatty acid is one of the most susceptible to oxidation reactions [43], due to its high degree of unsaturation and the positioning of its double bonds [44].

The results reported in this investigation coincide with what is reported by Böger et al. [45], who evaluated the oxidation of linoleic acid microencapsulated with gum arabic and maltodextrin by spray drying. The authors found that linoleic acid microencapsulated with gum arabic was more stable towards oxidation, but it depends on the percent of solids in the emulsion and the drying conditions used. In general, it can be seen that the microencapsulation of green coffee oil is beneficial for the protection of components such as fatty acids, which helps to extend the benefits they can bring and retards the oxidation process (Figure 4).

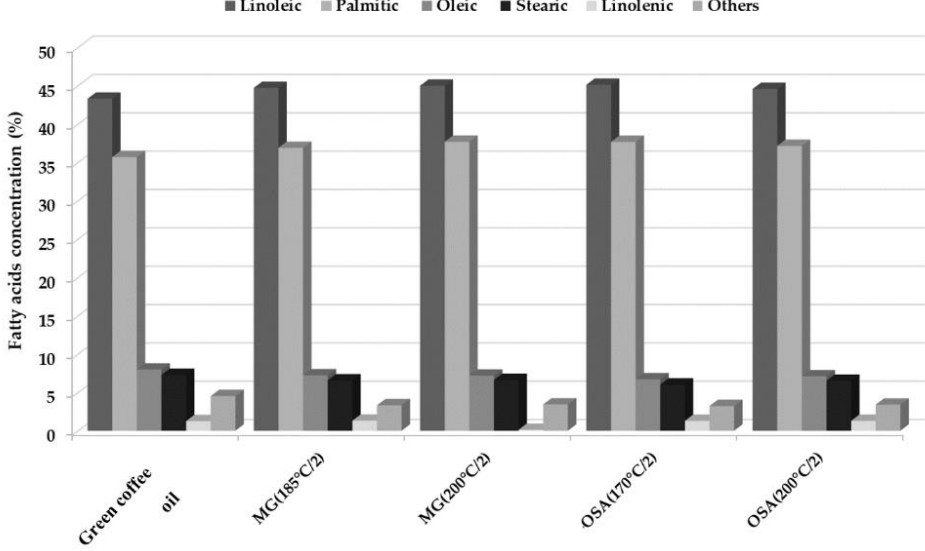

**Figure 4.** Fatty acid profile of the green coffee oil after one day of extraction, and microencapsulates with the higher peroxide index after 30 days of storage in 32% RH at 25 °C.

## 4. Conclusions

Monomodal droplet size green coffee oil emulsions were obtained by microfluidization using mesquite gum and OSA-starch as emulsifiers. The pressure and the number of passes of microfluidization reduced the mean droplet size, the polydispersity index, and zeta potential of the green coffee oil emulsions, promoting the physical stability by the electrostatic and steric mechanism. High microencapsulation efficiencies of the spray-dried emulsion were obtained for both wall-materials, indicating that the green coffee emulsion, temperature, and feed rate used in the spray drying process produced suitable microcapsules. The peroxide values of the microcapsules were lower than the bulk green coffee oil at similar storage conditions. Microcapsules produced with mesquite gum

showed a better protection of the green coffee oil than OSA-starch under accelerated storage conditions. Microencapsulation offers new opportunities in the development of nutraceutical or cosmetical products, improving the stability of lipids by increasing the shelf life of the green coffee oils retarding the apparition of the unpleasant tastes or odors.

**Author Contributions:** Conceptualization, H.E.-A., G.M.G.-M. and E.A.-G.; Formal analysis, H.E.-A., M.G.-V., G.M.G.-M. and E.A.-G.; Investigation, H.E.-A., M.G.-V., H.E.-S. and G.M.G.-M.; Methodology, M.G.-V. and H.E.-A.; Project administration, E.A.-G. and G.M.G.-M.; Resources, E.A.-G., G.M.G.-M. and H.E.-A.; Supervision, E.A.-G. and H.E.-A.; Writing—original draft, H.E.-A. and M.G.-V.; Writing—review and editing, H.E.-A., G.M.G.-M., H.E.-S. and E.A.-G.

**Funding:** This work was supported by the "SEP-CONACYT Investigation Basic" under Grant CB-2015-01-258118, and CONACYT-FORDECYT 292474; the Centro de Investigación y Asistencia en Tecnología y Diseño del Estado de Jalisco, A.C. (CIATEJ) and the Universidad de Guadalajara (UDG).

**Acknowledgments:** The authors would like to thank CONACYT for scholarship No. 717362 and projects CB-2015-01-258118 and FORDECYT 292474.

**Conflicts of Interest:** The authors declare no conflict of interest.

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
