# Peer review of "Oxidative Stability of Green Coffee Oil (Coffea arabica) Microencapsulated by Spray Drying"

_processes, doi:10.3390/pr7100734_

Round 1

Reviewer 1 Report

This paper have investigated the effect of conditions of emulsification and spray drying on the characteristics

of microcapsules.

I think that the following things should be described.

[1] why are Zeta potential of the Mesquite gum microcapsules not affected by pass number? 

[2] What conditions are the maximum yield of microcapsules ? 

Author Response

Reviewer 1

This paper have investigated the effect of conditions of emulsification and spray drying on the characteristics of microcapsules. I think that the following things should be described.

[1] why are Zeta potential of the Mesquite gum microcapsules not affected by pass number? 

Response: The negative zeta potential of the emulsions is related to the anionic nature of the carboxylic and hydroxylic groups of the mesquite gum and OSA-starch [26] and the dispersed medium conditions (pH, ionic strength and temperature) [23]. Mesquite gum is a suitable emulsifier and stabilizer agent since its protein fraction reduces the surface tension in a liquid-liquid interface. Besides, their glucuronic acids are dissociated forming a polyvalent macroion with a large number of counter-charge counterions, maximizing the repulsive forces between them [27]. The results showed that the emulsions produced with MG were not affected by the increase in the number of passes of microfluidization from one to five.

[2] What conditions are the maximum yield of microcapsules

Response: The encapsulation agent impacted the spray drying conditions that produce the highest yield of encapsulation. When mesquite gum was used as a wall material, an inlet temperature of 170°C and a feed rate of 2.5 mL/min produced a yield of encapsulation of 99.60 ± 0.10. On the other hand, when the OSA-starch was used, an inlet temperature of 200°C and a feed rate of 1.5 mL/min produced a yield of encapsulation of 99.0 ± 0.26

Reviewer 2 Report

Very well written, interesting, and potentially significant to the community. I enjoyed reading the manuscript. No suggestions for revisions

Author Response

The authors appreciate your comments.

Reviewer 3 Report

The work is an interesting and valuable contribution to understanding the effect of microfluidization and spray drying conditions on the physico-chemical properties of microencapsulated green coffee oil. The results obtained within the study might be useful for practical application. Generally, the results are consistent and properly discussed based on relevant references. The methods was clearly presented. This manuscript requires minor revision due to some shortcomings.

Comments that should be considered to make the manuscript suitable for publication:

L.16. Correct this phrase: “pharmaceutical o nutraceutical products”.

L.39. Consider replacing “their” by “its” if the “unique composition” regards oil. Otherwise, consider replacing “oil” by “oils” in Line 38.

L.105. What was the pressure of air supplied to the atomizer?

L.205 - 208. Consider replacing “their” by “its” as the described attributes in the following sentences concerns Mesquite gum and OSA-starch, respectively.

L.249. Replace “Presion” by “Pressure” in the description of horizontal axes.

L.288 - 289. How to explain the results demonstrating that the oxidation values of the microcapsules produced with OSA-starch were higher than the microcapsules of mesquite gum? In line 62 was stated that OSA-starches reveals resistance to oxidation. Although some explanation based on SEM images was provided but the reason of morphological changes caused by storage time were not explained sufficiently.

L.358. Replace “Percent” by “Fatty acids concentration” or other relevant term in the description of vertical axes.

L.369 - 378. Some practical recommendation which could be useful for industry application is missing.

Author Response

The work is an interesting and valuable contribution to understanding the effect of microfluidization and spray drying conditions on the physico-chemical properties of microencapsulated green coffee oil. The results obtained within the study might be useful for practical application. Generally, the results are consistent and properly discussed based on relevant references. The methods was clearly presented. This manuscript requires minor revision due to some shortcomings.

Comments that should be considered to make the manuscript suitable for publication:

L.16. Correct this phrase: “pharmaceutical o nutraceutical products”.

L.39. Consider replacing “their” by “its” if the “unique composition” regards oil. Otherwise, consider replacing “oil” by “oils” in Line 38.

L.205 - 208. Consider replacing “their” by “its” as the described attributes in the following sentences concerns Mesquite gum and OSA-starch, respectively.

L.249. Replace “Presion” by “Pressure” in the description of horizontal axes.

L.358. Replace “Percent” by “Fatty acids concentration” or other relevant term in the description of vertical axes.

Response. We appreciate the comments. Grammatical errors and suggestions were corrected throughout the document.

What was the pressure of air supplied to the atomizer?

Response. The pressure of air supplied to the atomizer was 0.4 MPa.

L.288 - 289. How to explain the results demonstrating that the oxidation values of the microcapsules produced with OSA-starch were higher than the microcapsules of mesquite gum? In line 62 was stated that OSA-starches reveals resistance to oxidation. Although some explanation based on SEM images was provided but the reason of morphological changes caused by storage time were not explained sufficiently.

Response. Several reports showed that the storage conditions impact the stability oxidation of spray-dried powders. Most of them used a relative humidity of 32% at 36 °C as accelerated storage conditions to observe the oil oxidation. Shamaei et al. [42] found that damages on the surfaces of microcapsules might contribute to the diffusion of oil droplets to the microcapsules surfaces.  In the part where it is mentioned that the OSA-starch microcapsules had higher values than those of GM, it was only intended to make the comparison between both biopolymers, however, that does not mean that the OSA-starch does not have that protective effect against oxidation

L.369 - 378. Some practical recommendation which could be useful for industry application is missing.

Response. The microencapsulation had a substantial effect as it protected green coffee oil, showing new opportunities for the development of nutraceutical foods or cosmetic formulations with adequate stability. In the case of food, it can improve organoleptic qualities and mask unpleasant tastes or odors, while in cosmetic formulations it acts as a protector against functional losses, increasing the effect of contained bioactive substances, however, the most studied dermatological application of green coffee oil, is certainly as a photoprotection aid.

Reviewer 4 Report

The research successfully evaluated the oxidative stability of microencapsulated green coffee oil by spray drying. The experimental design is well planned, the design and characterization of the particles was correct. I just suggest that the results of quantification of fatty acids should include a multifactor analysis, so that the influence of each treatment can be clearly established on the preservation of the fatty acids.This is due to the fact that Figure 6 does not represent the differences, and the error bars (representing standard deviation) are also missing. Finally, it would have been more complete, if the fatty acid profile had been followed over time, as was done with morphology, in the 30 days of storage.

Author Response

Response We appreciate the observations. A multifactorial analysis is not possible to perform it because of the fatty acid profiles of all samples were not obtained. The oxidative stability was used to select the most representative samples of the set, taking as criteria those that presented the lowest and highest values of the peroxide index. We offer an apology since the wording did not specify clearly that the image represents the profile of fatty acids after 30 days of storage.

Reviewer 5 Report

A manuscript entitled "Oxidative stability of green coffee oil (Coffea arabica) microencapsulated by spray drying" presents very interesting research.

In my opinion Authors need to consider the following points:

In the Section 2. not all reagents are listed.

In my opinion Authors should add table with name formulations, their composition and conditions of preparations (In the section 2.1. and 2.4).

Why 20% mesquite gum and OSA-starch were used to formulation of emulsion?

Which emulsion formulation was used to spray drying?

Author Response

In the Section 2 not all reagents are listed.

Response. We appreciate the observations, we already attended them in the text.

In my opinion, Authors should add table with name formulations, their composition and conditions of preparations (In the section 2.1. and 2.4).

Response We are very grateful for your opinion; however, it was considered that it was more appropriate to present the conditions in the results part since this data facilitated us that the discussion clearer. In addition, we did not assign names to the formulations and the only thing that differentiated the compositions was the use of mesquite gum or OSA-starch as wall material.

Why 20% mesquite gum and OSA-starch were used to formulation of emulsion?

Response: The wall materials used for microencapsulation of essential oils must be able to disperse at high concentrations and produce low viscosity dispersions. In general, the higher the solids content of the wall material, the greater the degree of protection. The concentration was selected based on recommendations from the ingredient supplier and the experience of working with these materials in the laboratory. Besides, several investigations conclude that the use between 10 and 30% of the wall material is suitable for spray drying. For example, Silva et al., 2014, “Influence of different combinations of wall materials and homogenization pressure on the microencapsulation of green coffee oil by spray drying”; Beristain and Vernon-Carter, 1994, “Utilization of mesquite (Prosopis juliflora) gum as emulsion stabilizing agent for spray-dried encapsulated orange peel oil”. Stefani Cortés-Camargo, et al. (2017) Microencapsulation by spray drying of lemon essential oil: Evaluation of mixtures of mesquite gum–nopal mucilage as new wall materials, Journal of Microencapsulation, 34:4, 395-407

Which emulsion formulation was used to spray drying?

Response Different conditions were used for each of the wall materials, for the mesquite gum: a pressure of 121.6 MPa and four passes of microfluidization; while for the OSA-starch a pressure of 206.8 MPa and three passes of microfluidization.